# The Therapeutic Effect of *Acanthopanax senticosus* Components on Radiation-Induced Brain Injury Based on the Pharmacokinetics and Neurotransmitters

**DOI:** 10.3390/molecules27031106

**Published:** 2022-02-07

**Authors:** Chen Song, Sijia Li, Fangyuan Duan, Mengyao Liu, Shan Shan, Ting Ju, Yingchun Zhang, Weihong Lu

**Affiliations:** 1Department of Food Science and Engineering, School of Chemistry and Chemical Engineering, Harbin Institute of Technology, Harbin 150001, China; songchen19891005@163.com (C.S.); lisijia201909@163.com (S.L.); 19B325016@stu.hit.edu.cn (F.D.); liumengyao621@gmail.com (M.L.); 15B925035@hit.edu.cn (S.S.); ab873460057@163.com (T.J.); zhangyingchun@hit.edu.cn (Y.Z.); 2School of Medicine and Health, Harbin Institute of Technology, Harbin 150001, China; 3National and Local Joint Engineering Laboratory for Synthesis, Transformation and Separation of Extreme Environmental Nutrients, Harbin Institute of Technology, Harbin 150001, China

**Keywords:** functional components of *Acanthopanax senticosus*, radiation, brain injury, neurotransmitters, pharmacokinetic

## Abstract

*Acanthopanax senticosus* (AS) is a medicinal and food homologous plant with many biological activities. In this research, we generated a brain injury model by ^60^Co -γ ray radiation at 4 Gy, and gavaged adult mice with the extract with AS, Acanthopanax senticocus polysaccharides (ASPS), flavones, syringin and eleutheroside E (EE) to explore the therapeutic effect and metabolic characteristics of AS on the brain injury. Behavioral tests and pathological experiments showed that the AS prevented the irradiated mice from learning and memory ability impairment and protected the neurons of irradiated mice. Meanwhile, the functional components of AS increased the antioxidant activity of irradiated mice. Furthermore, we found the changes of neurotransmitters, especially in the EE and syringin groups. Finally, distribution and pharmacokinetic analysis of AS showed that the functional components, especially EE, could exert their therapeutic effects in brain of irradiated mice. This lays a theoretical foundation for the further research on the treatment of radiation-induced brain injury by AS.

## 1. Introduction 

Radiation damage to the central nervous system has become an ongoing health challenge of recent decades largely due to the issues of radiation damage occurring duringhuman missions to other planets [1]. Cosmic space radiation can destroy biological macromolecules [2]. The most reported effects being observed after irradiation are altered cognitive function, manifesting via detriments in short-term memory, reduced motor function and behavioral changes [3]. Hence, it is required to search for more potent radioprotectors which can be easily administered.

*Acanthopanax senticosus* (AS) is a medicinal and food homologous plant used in China. Some pharmacological effects of AS, such as immune regulation, and properties, such as anti-radiation and anti-oxidation, have aroused researchers’ interest [4]. ASPS is one of the main active ingredients of AS in the nervous system. It can improve the symptoms of nerve defects in rats with cerebral ischemia-reperfusion injury and reduce the volume of cerebral infarction [5]. In terms of radiation protection, ASPS can reduce oxidative damage caused by radiation through its anti-infective activity [6]. Flavones compose the highest proportion of content of AS plants [7], which is a mixture that has the function of repairing heart damage and is used to treat ischemic heart disease [8]. Syringin and eleutheroside E are active ingredients isolated from AS. It has been proven that they have anti-fatigue, anti-oxidation, immune regulation and anti-inflammatory effects [9,10,11,12,13]. In addition, AS has also been shown to regulate the process of mediating the excitement and suppression of the central nervous system, and have a good therapeutic effect on related diseases that are caused by decreased learning and memory ability [14]. According to the special function of AS, it is important to understand the metabolic process of the bioactive components in AS to explore related natural medicine supplements. Meanwhile, the composition and pharmacological effects of components of AS are receiving more and more attention from scholars in various fields in recent years. In our previous article, we isolated the *Acanthopanax senticosus* polysaccharide (ASPS), flavones, syringin and eleutheroside E (EE) [7]. We proved that the AS could be a radioprotective agent against ion radiation-induced oxidative damage in animal tissues. In addition, we have demonstrated that ^60^Co-γ rays irradiation impaired learning, memory ability and hippocampal neurons in mice [15]. Therefore, in this research, we investigated the neuroprotective effects and metabolic characteristics of functional components in AS (ASPS, flavones, syringin, EE) on brain-injured mice.

## 2. Result and Discussion

### 2.1. Analysis of Behavioral Test

After radiation, all of the mice were handled with laboratory routine experiment-free exercise detection, which helped to ensure the health condition of the mice, in addition, the total food and water that was supplied was the same, and there was no significant difference in food and water left each day. The purpose of the water maze test is to detect the spatial discrimination ability and memory ability of mice [16] and the sucrose preference is to detect the anhedonia of mice [17].

Figure 1A clearly showed that all groups except for the control group were influenced by radiation, and compared to the model group, the number of snags in AS group (*p* < 0.01), flavones group (*p* < 0.05), syringin group (*p* < 0.01) and EE group (*p* < 0.05) significantly decreased. But the ASPS and Venlafaxine had no significant difference compared with the model group (*p* > 0.05). Meanwhile, All drug-treatment groups had a significantly reduced total time of arrival to the platform (Figure 1B, *p* < 0.01). It demonstrated that the functional components of *Acanthopanax senticosus* selected in this experiment can improve the spatial memory ability of brain-injured mice induced by radiation. Further more, the surcrose preference test reflected the anhedonia of mice. As can be seen from Figure 1D, the sugar partiality of the ASPS group, EE group, syringin group and positive control group was, respectively, 58.04%, 63.02%, 61.62% and 60.72%, which were extremely significant compared with the model group (*p* < 0.01).

Many reports have showed an altered behavioral performance in brain of mice exposed to radiation [18,19,20]. From the behavioral test, we know that the mice that were administered flavones, syringin and EE had improved spatial recognition and memory ability, while the groups administered syringin and EE had an improved anhedonia. We also found that the spatial discriminative ability, memory ability and anhedonia of the normal control group were significantly better compared to other groups, indicating that radiation had an irreversible effect on the behavior of mice. Behavioral results showed that syringin and EE had significant activities toward improving radiation-induced cognitive dysfunction in mice, similar to its effects found in previous studies investigating other cognitive dysfunction models caused by sleep interruption [21].

### 2.2. Analysis of Brain Histopathology Sections

Considering changes in cell morphology and that the number of hippocampal neurons is closely associated with learning and memory functions, the histological features of mice were determined by H&E staining. As we know, the behavior of animals is regulated by the central nervous system. The hippocampus is not only supposed to be associated with anxiety-like behavior and depression-like behavior [22,23], but also to play an important role in learning and memory [24]. As shown in Figure 2A, we observed a large number of neuron cells in hippocampus of mice in the control group, which were arranged neatly. However, in the radiation group, the number of neurons decreased and many pyknotic cells could be observed in hippocampus of mice. The hippocampi of treatment group mice showed less severe damage. Moreover, the brain tissue sections of the AS group and EE group had small cell cavities, abundant nerve cells, tight cell arrangements and normal morphology. It indicated that AS as a mixture is used to repair damage through the combination of various functional components, and EE as a functional component has a repairing effect on the damage of brain tissue. However, the ASPS group and flavonoid group had more morphologically abnormal nuclei than other treatment groups, indicating that there is no obvious effect in preventing hippocampal lesions. As the behavioral test showed, the ASPS and flavonoids were not significantly effective in improving learning and memory ability impairment induced by radiation.

### 2.3. The Functional Components of AS Increased the Antioxidant Enzyme Activity in the Brain of Irradiated Mice

Figure 3 showed the effect of AS on antioxidant activities in brain of irradiated mice. Compared with the control group, SOD and CAT activities were significantly decreased in the model group. Administration with the functional components of AS enhanced antioxidant activities, compared to the model group (*p* < 0.01). Among them, the AS extract and EE had a stronger antioxidant activity than other components. Although there is no obvious significant difference in the GSH activity between the control and model groups, administration with the functional components of AS increased the activity of GSH against the free radicals. The level of MDA was also considered as a maker of oxidative stress. Nevertheless, there was no significant difference in the MDA levels in our study. Many natural products including plant polysaccharides, animal polysaccharides and microbial polysaccharides have immune regulation, anti-tumor, anti-radiation, anti-inflammation, anti-fatigue and anti-aging effects, which are related to their antioxidant properties [25,26,27]. In recent years, the antioxidant properties of natural products such as flavonoids and saponins have also been extensively studied [28,29]. However, little is known about their effect on radiation-induced oxidative stress. What is more, oxidative stress may be one of the first events in neurodegenerative disease and is associated with neuronal damage and subsequent impaired spatial learning and memory [30]. In our study, the functional components of AS prevented radiation damage by enhancing the antioxidant capacity of brain. In summary, AS extract, flavonoids, ASPS and EE play an important role in the regulation of radiation-induced oxidative stress, protect nerve cells, and improve the oxidative damage caused by free radicals in brain tissue.

### 2.4. The Functinal Components of AS Changed the Level of Typical Neurotransmitters in Irradiated Mice 

Typical neurotransmitters are of great importance to learning and memory ability [31]. Data describing the impact of AS composition on the levels of brain monoamine neurotransmitters in irradiated mice are illustrated in Figure 4. After irradiation by ^60^Co-γ rays, the level of NE in the model group significantly decreased compared to the control group (*p* < 0.05 Figure 3A). What’s more, the level of NE in the ASPS and flavones groups showed a significant increase compared with the model group (*p* < 0.05), but other groups had no significant differences compared with the model group. Similarly, the level of 5-HT had the same trend between the control group and model group (*p* < 0.01, Figure 4B). Administration with EE and venlafaxin to the irradiated mice increased the level of 5-HT (Figure 4B). Then we detected the contents of MAO to evaluate the degree of brain damage. The results showed that ^60^Co-γ rays irradiation increased the level of MAO. Fortunately, EE, syringin and venlafaxin could significantly prevent MAO from increasing (*p* < 0.05, Figure 4C). Acetylcholine acts as a neurotransmitter in the central nervous system and is involved in hippocampus-mediated memory function. There is a huge significant difference in the level of ACH in brain tissue between the control group and model group (*p* < 0.01, Figure 4D). In the AS group, syringin group and EE group, the level of ACH increased compared to the model group (*p* < 0.05, Figure 4D). NE is not only a neurotransmitter synthesized and secreted by nerve endings, but also a hormone synthesized and secreted by the adrenal glands. It is a marker metabolite of brain injury in mice. Studies have shown that reduced NE levels can cause neurotoxicity and pro-inflammatory decrease [32]. The lack of NE will reduce the body’s nerve excitability, which plays an important role in the body’s central system activities such as learning and memory. Serotonin (5-HT) is an excitatory neurotransmitter that is highly abundant in the cerebral cortex and nerve processes and is a hallmark of nerve damage. It was reported that lesioning of serotonergic afferents attenuates memory deficits produced by cholinergic lesions [32]. Previous studies have demonstrated that rutin is an acetylcholinesterase (AChE) inhibitor in human plasma in vitro [33], suggesting that rutin may be valuable in treatments where the inhibition of AChE is employed, including neurological disorders. The ACh in the hippocampus was decreased after radiation exposure. This abnormal state can be reversed by the administration of flavones, syringin and EE. Here, our results suggested that flavones, syringin and EE may be potential therapeutic agents against radiation-induced cognitive dysfunction. The possible mechanism can be attributed to neuroprotection via AChT enhancement, which promotes the synthesis of ACh. Monoamine oxidase (MAO) is an important indicator to measure brain tissue damage. There are two subtypes, MAO-A and MAO-B, which can catalyze the biogenic amines produced by various life activities. On the other hand, biogenic amines also play an important role in normal life activity for the body. Monoamine oxidase can selectively decompose 5-HT, NE and other neurotransmitters closely related to learning and memory ability. As previous research has shown, excessive monoamine oxidase can cause Parkinson’s syndrome [34]. The higher level of MAO indicated more severe brain damage. In our results, we found the EE and syringin decreased the level of MAO to prevent brain damage.

### 2.5. Distribution of Functional Components of Acanthopanax Senticosus in the Different Tissues

In order to study the distribution of the functional components of *Acanthopanax senticosus* in various tissues, we calculated the relative metabolism of the functional components in the tissues after 24 h of administration. As shown in Table 1, the functional components of *Acanthopanax senticosus* can be detected in the liver, thymus, spleen, kidney, testes and brain, with significant differences. The content of ASPS in the liver is relatively high, and studies have found that ASPS can effectively reverse liver and kidney damage [35]. Appendix A shows the same trend. The levels of flavonoids of AS were high in liver, spleen, kidney, heart and brain tissue. Studies have shown that rutin had high antioxidant activity, and can effectively treat cardiovascular and cerebrovascular diseases [36,37]. Therefore, we speculate that flavonoids of AS can play a role in improving the oxidative damage caused by radiation in mice. The levels of syringin and EE were high in liver and spleen, where they accumulated and metabolized. Syringin and EE are fat-soluble, which means they can more easily penetrate biofilms to exert their medicinal effects in organs. EE is known to reduce physical fatigue, is anti-inflammation and inhibits high glucose [38,39]. Meanwhile, in the aforementioned results, EE would play a significant role in preventing the behavioral impairment of radiated mice and regulating the level of neurotransmitters. It is speculated that EE can improve the learning and memory ability of radiated mice by regulating neurotransmitters in brain. Studies have shown that a combination of syringin and tilianin effectively exerted antidiabetic effects and improved cardiac function [40]. Syringin has an ability to raise the release of ACh from nerve terminals, which in turn stimulates muscarinic M3 receptors in pancreatic cells and augments the insulin release to result in a plasma glucose lowering action [41]. In addition, according to the Appendix A, the bioavailability of EE and syringin in brain tissues is relatively higher than other components. Generally, for some natural products, such as lycium barbarum polysaccharides and jujube polysaccharides, their immune activity and antioxidant effect have been widely studied and confirmed [42]. Similarly, ginsenosides and green tea polyphenols, which have been studied more in recent years, have also been proved to have strong immunomodulatory activities and anti-tumor effects [43,44,45]. The main difference between the functional components of *Acanthopanax senticosus* in this study and those above is that they focus on radiation-induced damage and play an important role in nerve and oxidative stress. We systematically studied the distribution of each component in vivo, which laid a theoretical foundation for its further application.

Since nerves are sensitive to radiation, we focused on the effects of functional components of AS on brain injury. Next, we analyzed the pharmacokinetics of functional components of AS in brain. 

### 2.6. Pharmacokinetic Study of the Functional Composition of Acanthopanax Senticosus in Brains of Irradiated Mice

Pharmacokinetics is an important indicator for studying the metabolic regularity of drugs in vivo. Furthermore, the change trend of the pharmacokinetic curve can reflect the change of concentration, and pharmacokinetic parameters can quantify the drug metabolism in the form of numerical values. In some cases, the durability of the drug’s effect can be obtained by calculating the half-life of the drug, and reasonable recommendations can be made for the dosage. From the results (Figure 5), the pharmacokinetic curve showed a trend of rising first and then falling, indicating that the time point selected in the experiment fully expressed the absorption and metabolism of polysaccharides, flavones, syringin and EE, reflecting the characteristics of pharmacokinetics. It is worth noting that there were double peaks in the pharmacokinetic curve of flavones, which indicating that flavones may be converted into other monomer substances to play a role in repairing brain injuries induced by radiation. It is also necessary to analyze the specific components of flavones in the brain tissue by mass spectrometry. 

Table 2 shows the pharmacokinetic parameters of functional components of Acanacanax in the brain. The Tmax of functional components of AS is the same, all of which are 2 h, indicating that 2 h is the optimal time point for drug absorption. Besides, the half-lives of EE and syringin were significantly longer than the polysaccharide and flavones (*p* < 0.05), which indicated that the saponins have a long-lasting effect. From the results of bioavailability, it could be seen that the bioavailability of EE was significantly higher than other components. In the chemical structure formula of EE, the benzene ring contains -OH, which can bind with free radicals and inhibit oxidative damage. We speculated that EE could improve the radiation injury of brain tissue by scavenging free radicals. Pharmacokinetic and tissue distribution studies all proved that EE played a major role in the brains of irradiated mice, which provided a new idea and research basis for subsequent studies on Acanthopanax Senticosus.

## 3. Materials and Methods

### 3.1. Experimental Material

The Chinese herbal medicine Ancathopanax Senticosus (AS) was collected from the lesser khingan mountain area of Heilongjiang Province. The 4-year-old AS was randomly selected. The roots and leaves were separated and stored in a refrigerator at −80 °C. A total of 250 g of roots and leaves were ground separately and extracted with distilled water at 65 °C for 3 h. The above operation was repeated 3 times to mix all the decoctions. Then, all the mixed decoctions were filtered with gauze and concentrated with rotary evaporator to obtain 50 g of AS extract [46]. 

ASPS and flavones of AS were purchased from Xi’an Shengqing Biotechnology Co., Ltd. (Xi’an, China). The content of ASPS of AS was 95% (calculated by glucose) and the content of total flavones was 90% (calculated by rutin). The syringin and (EE) were purchased from Nanjing Xinhou Biotechnology Co., Ltd. (Nanjing, China; molecular formula C_58_H_92_O_25_ and C_34_H_46_O_18_, molecular weight 1188.59 and 742.72). The purity of syringin and eleutheroside E (EE) were both 98% detected by HPLC.

### 3.2. Animal Experiments

Adult male Kunming mice, 6–8 weeks old, weighing 20 ± 2 g were obtained from the Animal Experimental Center of the 2nd Affiliated Hospital of Harbin Medical University (Harbin, Heilongjiang, China). The certification number was SCXK-(HEI)2019-001. All of the animal experimental procedures followed the National Institutes of Health guidelines for the care and use of laboratory animals, and were evaluated and approved by local ethics committee of the Harbin Institute of Technology. All animals were maintained in an environmentally controlled breeding room with a regular 12 h light cycle at 22 ± 2 °C. The mice were allowed at least 1 week to adapt to the environment before being used for experiments.

All mice were assigned to use in different experiments and divided into two large groups for experiments: group 1—behavioral experiments, histological experiment, neurotransmitter content and the distribution of the functional components in different tissues; group 2—pharmacokinetic study in brain tissue of ASPS, flavones, syringin and EE.

Referring to the Chinese Pharmacopoeia 2010 edition of the medication regulations, combined with the “Health food functional evaluation procedures and test methods” requirements for experimental animals and venlafaxine drug-use dose instructions, the experimental groups and drug intake were as follows. For group 1, mice were randomly divided into 8 groups (*n* = 8 per group): the normal control group (Control); the model group (Model); the AS extract group (AS, dosage: 235.7 mg/kg/d); the ASPS group (ASPS, dosage: 2.09 mg/kg/d); the flavones group (Flavones, dosage: 3.9 mg/kg/d); the EE group (EE, dosage: 204.6 μg/kg/d); the syringin group (Syringin, doage: 242 μg/kg/d); and the positive control group (Venlafaxine, dosage: 13.75 mg/kg/day). The mice were intragastrically administered the drugs once a day at 8:00 am. All the active ingredients were dissolved in normal saline; the normal control group and model group were administered the same amount of normal saline, and mice were continuously intragastrically administered for 2 weeks (14 days). Except for the normal control group, other groups of mice were irradiated with ^60^Co-γ rays of, 4 Gy, and 0.9 Gy/min. 

Group 2: mice were divided into 6 small groups (*n* = 6, per group), the normal control group (Control); the model group (Model); the ASPS group (ASPS, dosage: 2.09 mg/kg/d); the flavones group (Flavones, dosage:3.9 mg/kg/d); the syringin group (Syringin, dosage: 242 μg/kg/d); and the EE group (EE, dosage: 204.6 μg/kg/d). After adaptive feeding for 7 days, except for the normal control group, other groups of mice were irradiated with the irradiation conditions, the same as in group 1. After 24 h of irradiation, the brain tissue samples were collected from the mice at zero, 0.5 h, 1 h, 2 h, 4 h, 8 h, 12 h, 16 h and 24 h after euthanizing. 

### 3.3. Behavioral Test

#### 3.3.1. Water Maze Test

Experimental methods and principles have been described in the previous articles [47]. Briefly, the mice of group 1 were placed into the pool from the head wall. The time of the mice to find the platform was recorded, and the mouse labyrinth swimming condition was observed, with a comprehensive evaluation of the spatial discrimination, learning and memory abilities, which were reflected by using the number of collisions and the time required to reach the platform. Before the formal test, all the mice were trained for 5 days, and then tested on the 6th day.

#### 3.3.2. Sugar Partiality Experiment

Before the test, two bottles of sucrose water were provided to the mice for 24 h, then changed into a bottle of sucrose water and a bottle of distilled water for 24 h. After 24 h of water deprivation, the distilled water and sucrose water were provided to the mice for 4 h and the location of the bottle was changed during the test to avoid the effect of memory. Sucrose partiality was defined as the ratio of the volume of sucrose vs. water consumed during the test and using the equation: Sucrose preference % = (Vsucrose/Vsucrose + Vwater) × 100, where Vsucrose is the volume of sucrose consumption, and Vsucrose + Vwater is summation volume of sucrose consumption and water consumption.

### 3.4. Hematoxylin and Eosin Staining

Brain tissues of all examined groups were fixed in 4% polyformaldehyde for 4–6 h. Then, the specimens were washed and dehydrated in ascending grades of ethyl alcohol, cleared in xylene and embedded in paraffin wax. Sections of 5 μm in thickness were cut out, deparaffinized and stained with hematoxylin and eosin (H&E) for examination under the light microscope.

### 3.5. Measurement of Antioxidant Enzyme Activity in Brain Tissue

Malondialdehyde (MDA) levels, glutathione peroxidase (GSH- Px) activities, catalase (CAT) and superoxide dismutase (SOD) activities in the liver were all determined spectrophotometrically using commercially available assay kits (Nanjing Jiancheng Bioengineering Institute, Nanjing, China).

### 3.6. Neurochemical Determinations

The neurotransmitter content associated with learning and memory ability in mice was determined by ELISA kits (Shanghai Lengton Bioscience Co., LTD, Shanghai, China), including acetylcholine (Ach), norepinephrine (NE) and 5-hydroxytryptamine (5-HT) in brain tissues. The brain tissues were accurately weighed, cut into slices and homogenated after adding the appropriate amount of normal saline (1 mL/0.1 g tissue). The tissue homogenate was centrifuged at 1000× *g* for 5 min after vortexing for 3 min. The supernatant (1.0 mL) was placed into another tube and recentrifuged at 3000 g for 10 min. The supernatant was used for neurochemical determination according to the commercial kit’s instruction. 

### 3.7. Distribution and Pharmacokinetic Analysis of AS Components in the Brain Tissues of Irradiated Mice 

The brain tissue of mice was taken out, and it was made into a tissue homogenate with an appropriate amount of normal saline.

#### 3.7.1. Determination of the Content of ASPS and Flavones

The content of ASPS was determined by phenol sulfuric acid method. The OD value of the sample was measured by a microplate reader at a wavelength of 490 nm, using D-glucose as the standard, the calibration curve (y = 0.1903x + 0.0269 R^2^ = 0.9997). The content of flavonoids in AS was determined by aluminum nitrate colorimetry, and rutin was used as the standard. The content of flavonoids was determined by the aluminum nitrate colorimetry, and the standard curve was made with rutin as the standard. Briefly, 0.3 mL of 5% sodium nitrite solution was added to 1 mL of the sample, left for 6 min, then 0.3 mL of 10% aluminum nitrate solution was added, and after standing for 6 min, 4 mL of 1 mol/L sodium hydroxide solution was added, and finally diluted to 10 mL with 30% ethanol, shaken well, and left to stand for 10 min. The OD value of the sample was measured by an ultraviolet spectrophotometer at a wavelength of 510 nm. The standard equation was Y = 0.0119X + 0.0064 (R^2^ = 0.9998). 

#### 3.7.2. Determination of the Content of Syringin and EE

The syringin and EE in the homogenate of animal tissue were vortexed with methanol, centrifuged at 3000 rpm/min for 10 min and filtered through a 0.45 um microporous membrane to obtain a solution for determination. A Thermo U3000 series liquid chromatograph (Thermo Fisher), equipped with a quaternary gradient pump and a UV detector was used. A HPLC method was developed using a reversed-phase C18 column (Agilent-TC, 250 mm, 4.6 mm, 5 μm i.d.) with the column temperature at 30℃. Sample injection quantity was 10 μL; the elution solvent consisted of water, with 0.1% phosphoric acid (A) and acetonitrile (B) with the following gradient program: 0–10 min, 90%A; 10-20 min, 85%A; 20–30 min, 90%A. The flow rate was kept at 1 mL/min, and the absorbance was measured at a wavelength of 220 nm. The standard curve of the two active substances was obtained according to the content of the corresponding standard—the area under the peak, and the content of syringin and EE in each tissue was obtained according to the peak areas of syringin and EE in each tissue.

#### 3.7.3. Distribution of Functional Components of *Acanthopanax Senticosus* in Different Organs of Mice

All the mice were administrated with the functional components of *Acanthopanax Senticosus* for 14 days, the contents of the ASPS and flavones were determined according to 2.6.1 and the contents of syringin and EE were detected according to 2.6.2 after 24 h since the last administration. 

#### 3.7.4. Pharmacokinetic Analysis

Tissue homogenate level–time profiles of ASPS, flavones, syringing and EE concentration were assembled by drawing a curve between average tissue homogenate levels (in ng/mL) and time (in hours). The parameters were: area under the curve to 24 h area under the curve to infinity (AUC_inf_); maximum concentration of drug in serum (*C*_max_); time to achieve *C*_max_ (*T*_max_); and half-life (*t*_1/2_). The average residence time (MRT), clearance (CL) and volume of distribution at steady state (Vss) were calculated by repeated calculations using PK solver 2.0.

### 3.8. Statistical Analysis

All the values were given as mean standard deviation. All of the results were analyzed by one-way analysis of variance (ANOVA) using SPSS 19.0. Pharmacokinetics data were evaluated with PK solver 2.0. A *p*-value < 0.05 indicated that there was a significant difference. 

## 4. Conclusions

In summary, the main purpose of this work was to investigate the repairing effect of AS components on radiation-induced brain injury in mice. The functional components of Acanthopanax senticoucus can effectively prevent brain damage caused by radiation, and the effect of EE is the most significant. While improving learning and memory, they also altered the levels of neurotransmitters in the irradiated mice. These results provided useful information for further study of its pharmacological actions in the treatment of nervous system disorder, which would open windows of opportunity for further research on the mechanisms of AS in the treatment of radiation-induced brain injury. Though all the evidence in support of potential therapeutic efficacy of eleutheroside E in brain damage induced by radiation, further studies of eleutheroside E for the therapeutic targets and the signaling pathways are required.

## Figures and Tables

**Figure 1 molecules-27-01106-f001:**
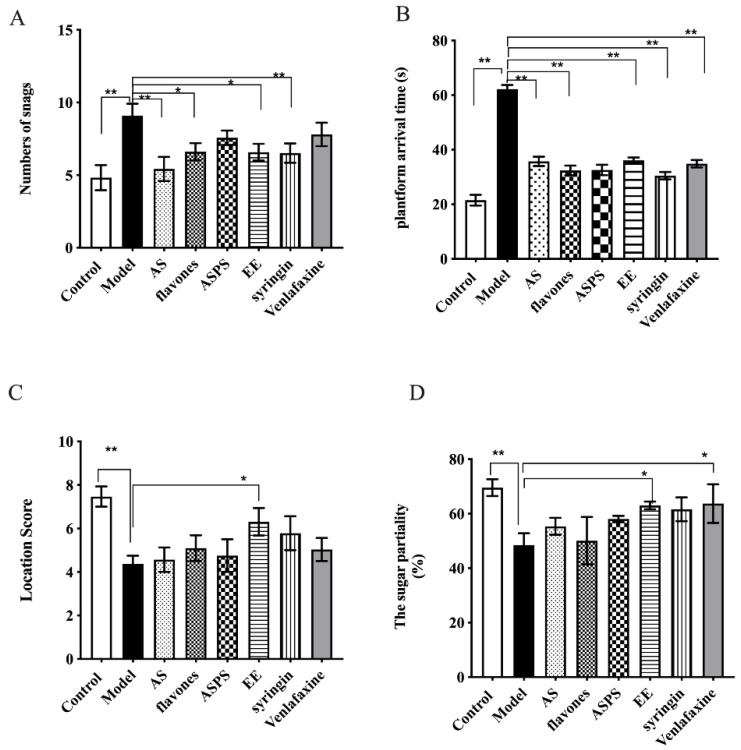
The completion of water maze in each group of mice. (**A**) The number of snags of each group; (**B**) the platform arrival time; (**C**) the location score; (**D**) the sugar partiality of each group (*n* = 8 per group, * *p* < 0.5, ** *p* < 0.01, significant differences between the groups were tested by one-way analysis of variance).

**Figure 2 molecules-27-01106-f002:**
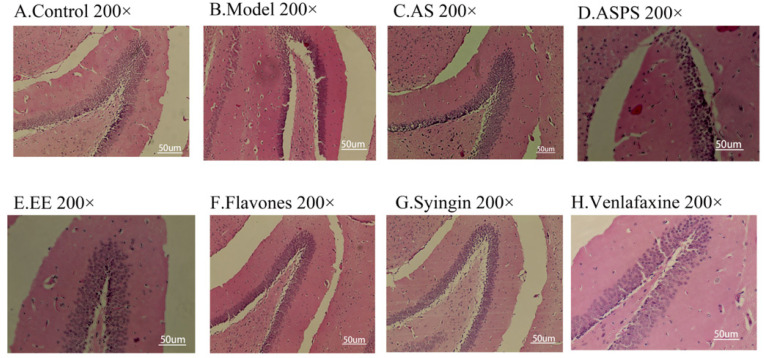
Histopathological sections of hippocampus in different groups of mice: (**A**) Control group; (**B**) Model group; (**C**) AS group; (**D**) ASPS group; (**E**) EE group; (**F**) Flavones group; (**G**) Syringin group; (**H**) Venlafaxine group. (Three mice were used in this analysis and the representative pictures were shown, scale bar = 50 um).

**Figure 3 molecules-27-01106-f003:**
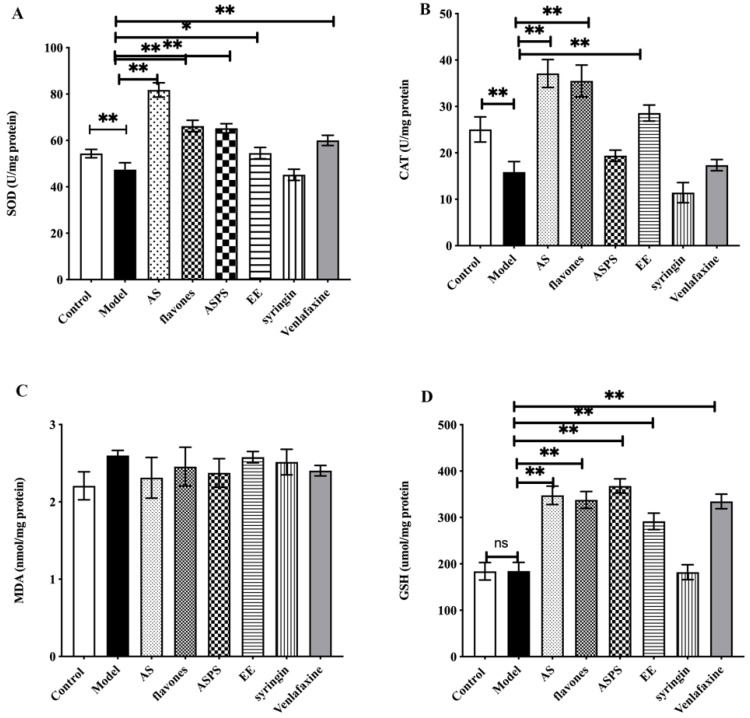
The level of antioxidant enzyme activity in brain of each group. (**A**) The level of SOD in the brains of mice; (**B**) the level of CAT in the brain of mice; (**C**) the level of MDA in the brain of mice; (**D**) The level of GSH in the brain of mice. (* *p* < 0.05, ** *p* < 0.01, *n* = 5, significant differences between the groups were tested by one-way analysis of variance.)

**Figure 4 molecules-27-01106-f004:**
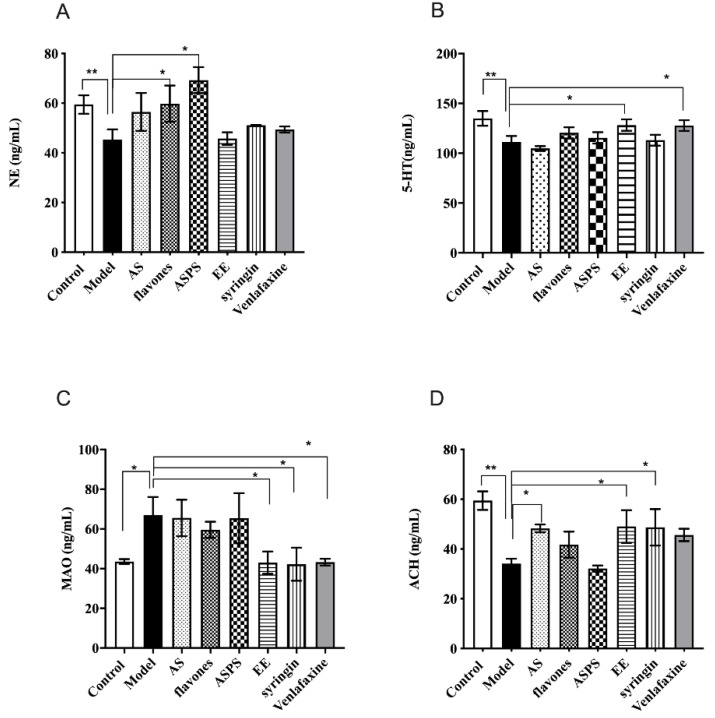
The level of neurotransmitters in brain of each group. (* *p* < 0.05, ** *p* < 0.01, *n* = 5). (**A**) The level of NE in the brains of mice; (**B**) the level of 5-HT in the brains of mice; (**C**) the level of MAO in the brain of mice; (**D**) the level of ACH in brain of mice. (* *p* < 0.05, ** *p* < 0.01, *n* = 5, significant differences between the groups were tested by one-way analysis of variance.)

**Figure 5 molecules-27-01106-f005:**
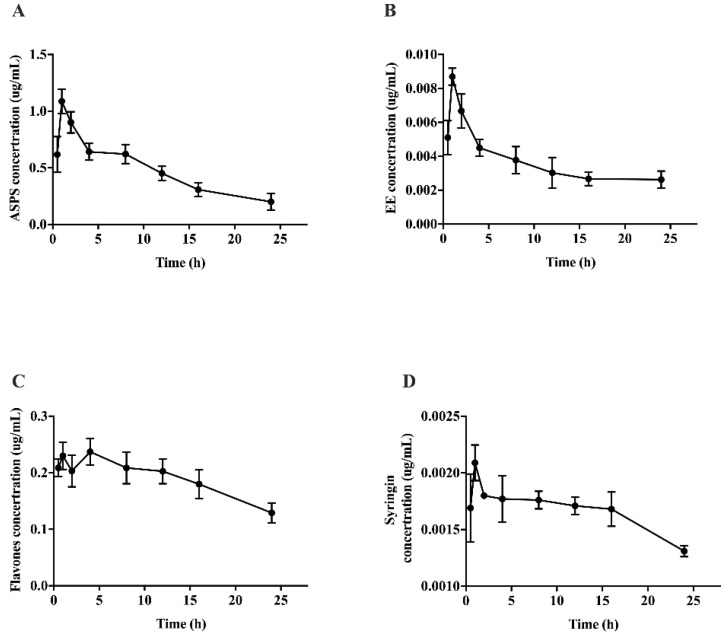
Pharmacokinetic curve of functional components of Acanthopanax senticocus in brain tissue. (**A**) Pharmacokinetic curve of ASPS; (**B**) pharmacokinetic curve of EE; (**C**) pharmacokinetic curve of flavones; (**D**) pharmacokinetic curve of syingin.

**Table 1 molecules-27-01106-t001:** The content of functional components of AS in tissues (ug/mL).

	Polysaccharide (μg/mL)	Flavones (μg/mL)	Syringin (μg/mL)	EE (μg/mL)
Liver	1.88 ± 0.06 ^a^	10.62 ± 0.05 ^d^	0.64 ± 0.01 ^a^	1.06 ± 0.01 ^a^
Thymus	0.62 ± 0.04 ^c^	6.84 ± 0.05 ^e^	0.14 ± 0.00 ^b^	0.08 ± 0.01 ^b^
Spleen	0.64 ± 0.01 ^b^	12.07 ± 0.02 ^c^	0.36 ± 0.01 ^b^	1.42 ± 0.03 ^a^
Kidney	0.72 ± 0.03 ^bc^	13.77 ± 0.03 ^b^	0.72 ± 0.03 ^a^	1.02 ± 0.01 ^a^
Testis	0.52 ± 0.03 ^c^	6.31 ± 0.01 ^f^	0.02 ± 0.001 ^c^	0.12 ± 0.01 ^b^
Heart	0.91 ± 0.04 ^c^	15.07 ± 0.10 ^a^	0.06 ± 0.02 ^c^	0.02 ± 0.01 ^b^
Brain	0.07 ± 0.25 ^d^	12.07 ± 0.09 ^c^	0.04 ± 0.01 ^d^	0.05 ± 0.09 ^b^

Vertical comparison of different letters in the upper right corner means significant difference (*p* < 0.05, significant differences between the groups were tested by one-way analysis of variance).

**Table 2 molecules-27-01106-t002:** Pharmacokinetics parameters of AS component in brain.

	Polysaccharide	Flavones	Syringin	EE
t1/2 (h)	16.11 ± 0.99 ^b^	7.92 ± 0.34 ^c^	38.58 ± 0.84 ^a^	38.35 ± 0.34 ^a^
Tmax (h)	2.00 ± 0.00 ^a^	2.00 ± 0.00 ^a^	2.00 ± 0.00 ^a^	2.00 ± 0.00 ^a^
CL (mg mL/μgh)	0.12 ± 0.03 ^b^	0.22 ± 0.06 ^b^	0.10 ± 0.08 ^b^	0.91 ± 0.06 ^a^
Cmax (μg/mL)	38.12 ± 0.08 ^a^	3.31 ± 0.03 ^c^	1.26 ± 0.09 ^d^	11.43 ± 0.03 ^b^
AUC(0-inf) (μgh/mL)	16.70 ± 1.84 ^b^	17.65 ± 1.01 ^c^	0.23 ± 0.11 ^d^	0.22 ± 0.05 ^a^
F (%)	0.04 ± 0.01 ^b^	0.10 ± 0.05 ^c^	0.08 ± 0.03 ^d^	0.15 ± 0.02 ^a^
MRT (h)	21.87 ± 1.95 ^b^	84.28 ± 0.44 ^a^	80.28 ± 0.14 ^a^	53.05 ± 0.44 ^c^
V (mg mL/μg)	0.31 ± 0.07 ^c^	1.15 ± 0.14 ^b^	1.48 ± 0.54 ^b^	3.71 ± 0.14 ^a^

Vertical comparison of different letters means significant difference (*p* < 0.05, significant differences between the groups were tested by one-way analysis of variance).

## Data Availability

The data that support the findings of this study are available from the corresponding author on reasonable request.

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
