# Peer review of "The Therapeutic Effect of Acanthopanax senticosus Components on Radiation-Induced Brain Injury Based on the Pharmacokinetics and Neurotransmitters"

_molecules, 2022, doi:10.3390/molecules27031106_

Round 1
Reviewer 1 Report
This study systematically investigated the protective effects of different functional components of Acanthopanax senticosus on brain injury induced by simulated space radiation in mice. The pharmacokinetics of each component and its metabolism in irradiated mice were clearly stated in this paper. At the same time, the protective effects of each component on learning and memory impairment of irradiated mice were analyzed from the aspects of behavior, oxidative damage and neurotransmitter.The research is feasible and of scientific significance.
Overall, the article is well organized and its presentation is good. However, some minor issues still need to be improved.
(1) For the section of 3.3, the discussion of the results needs to be strengthened.
(2) For the figure legends, some details needed to be add,such as describing the performed statistics.
(3) Check the grammar of the whole text and further improve the language editing.
Author Response
Dear editor and reviewers:
Thank you very much for your constructive suggestions with regard to our paper " The Therapeutic Effect of Acanthopanax Senticosus Components on Radiation Induced Brain Injury Based on The Pharmacokinetics and Neurotransmitters. We express our sincere gratitude to the editors’ and reviewers’ conscientious work in the whole process, your comments and/or suggestions are so important for the improvement of our manuscript. We have tried our best to revise and improve the manuscript according to the reviewers’ good comments. The contents related to the reviewers’ comments and revisions are marked in red throughout the revised manuscript. Responses to the reviewers’ comments are in the attachment. The revised manuscript was marked up using the “Track Changes” function of MS word.
Please see the attachment.

Reviewer 2 Report
Interesting subject but needs some explanations and improvement.
Line 39 and line 43:capital letters after a comma, should it be a dot?
Line 56: why 60Co-γ rays irradiation? What is the importance of this radiation to animals or humans?
Line 63: who certified the collected plant?
Line 67: says «mixing the boiling liquid», water at 45 oC, boils?
Line 68: says «purify the resin? Which resin?
Description from line 66-68 is confusing.
Line 69: «ASPS and flavones were purchased from Xi'an Shengqing Biotechnology Co., Ltd» which flavones?
Line 70: «The content of ASPS was 95% and the content of flavones was 90%, syringin and EE were 70 purchased from Nanjing Xinhou Biotechnology Co., Ltd.» How were these contents determined? This sentence does not make sense. Probably a dot before syringin.
Line 72-73: The purity was 98% detected by HPLC respectively. The purity of what, Syringin? Respectively? The sentence should be rewritten.
Line 92: venlafaxine drug is used for? It should be indicated
Why the control groups were not irradiated? There should be a comparison with irradiated animals that were not receive the compounds, to see the effect of the extracts and chemicals.
Line 152: «the measurement of the AS component penetrated the blood-brain barrier was performed». How was the brain extract prepared for these determinations? The methods are not indicated.
Line 157: «The content of flavonoids in AS was determined by colorimetry, and rutin was used as the standard». Which colorimetric method?
Line 163: «organic filter» which?
Figure 3 legend should have explanation about the figures a-d.
Figure 4 legend should have the explanations for each sub-figure.
Line 358: in vivo in italic.
Table 1 title should have the indication of the values units, that is ug/g of what? Organ?
General: how much of each component is in the extract given to the animals; How much of what was given to the animals is spread over the organs, (%)?
In the Mat and Method section says that flavonoids, polysaccharides, EE and syringin were quantified in the brain, but in the extract weren’t they quantified? How much of each component is given to the animals?
The results should be discussed. Most of them are only the authors results without discussion and comparison with other or similar compounds acting for the same objective.
Author Response
Dear editor and reviewers:
Thank you very much for your constructive suggestions with regard to our paper " The Therapeutic Effect of Acanthopanax Senticosus Components on Radiation Induced Brain Injury Based on The Pharmacokinetics and Neurotransmitters. We express our sincere gratitude to the editors’ and reviewers’ conscientious work in the whole process, your comments and/or suggestions are so important for the improvement of our manuscript. We have tried our best to revise and improve the manuscript according to the reviewers’ good comments. The contents related to the reviewers’ comments and revisions are marked in red throughout the revised manuscript. Responses to the reviewers’ comments are in the attachment. The revised manuscript was marked up using the “Track Changes” function of MS word.
Please see the attachment

Round 2
Reviewer 2 Report
Thanks for improving the work. It is fine now.